# Internet-delivered attentional bias modification training (iABMT) for the management of chronic musculoskeletal pain: a protocol for a randomised controlled trial

Christina Liossi ,[1] Tsampikos Georgallis,[1] Jin Zhang,[1] Fiona Hamilton,[1] Paul White,[2] Daniel Eric Schoth[1]

[1]Pain Research Laboratory, School of Psychology, University of Southampton, Southampton, Hampshire, UK
[2]Applied Statistics Group, Engineering, Design and Mathematics, University of the West of England, Bristol, UK

**Correspondence to**
Professor Christina Liossi;
cliossi@soton.ac.uk

## ABSTRACT

**Introduction** Chronic musculoskeletal pain is a complex medical condition that can significantly impact quality of life. Patients with chronic pain demonstrate attentional biases towards pain-related information. The therapeutic benefits of modifying attentional biases by implicitly training attention away from pain-related information towards neutral information have been supported in a small number of published studies. Limited research however has explored the efficacy of modifying pain-related biases via the internet. This protocol describes a randomised, double-blind, internet-delivered attentional bias modification intervention, aimed to evaluate the efficacy of the intervention on reducing pain interference. Secondary outcomes are pain intensity, state and trait anxiety, depression, pain-related fear, and sleep impairment. This study will also explore the effects of training intensity on these outcomes, along with participants' perceptions about the therapy.

**Methods and analysis** The study is a double-blind, randomised controlled trial with four arms exploring the efficacy of online attentional bias modification training versus placebo training theorised to offer no specific therapeutic benefit. Participants with chronic musculoskeletal pain will be randomised to one of four groups: (1) 10-session attentional modification group; (2) 10-session placebo training group; (3) 18-session attentional modification group; or (4) 18-session placebo training group. In the attentional modification groups, the probe-classification version of the visual-probe task will be used to implicitly train attention away from threatening information towards neutral information. Following the intervention, participants will complete a short interview exploring their perceptions about the online training. In addition, a subgroup analysis for participants aged 16–24 and 25–60 will be undertaken.

**Ethics and dissemination** This study has been approved by the University of Southampton Research Ethics Committee. Results will be published in peer-reviewed journals, academic conferences, and in lay reports for pain charities and patient support groups.

**Trial registration number** NCT02232100; Pre-results.

### Strengths and limitations of this study

► This protocol describes an internet-delivered intervention to explore the efficacy of attentional bias modification training in patients with chronic musculoskeletal pain.

► The study will recruit a diverse sample of participants in terms of age, socioeconomic status and geographical region.

► Results will provide important information regarding dose effects, age effects, and participants' experiences and attitudes using the online intervention.

► There is limited control over the conditions under which training is completed.

► Independently verifying participant diagnosis is impossible.

## INTRODUCTION

Chronic musculoskeletal pain (CMSK) is pain arising from the bones, muscles, ligaments, tendons and/or joints lasting for more than 3 months.[1] Prevalence estimates of CMSK vary considerably, affecting between 11.4% and 24% of the adult population[2] and between 4% and 40% of children and adolescents.[3] Negative effects on quality of life are commonly reported in adult[4–7] and paediatric[8–11] CMSK populations, and theory-driven research[12–14] has shown that patients with chronic pain exhibit attentional biases (ie, a selective attention) for salient pain-related words (eg, refs [15–18]) and images (eg, refs [19 20]) relative to neutral information. Three meta-analyses have supported the presence of attentional biases in patients with chronic pain,[21–23] although evidence for the time course of such biases is mixed. Specifically, two meta-analyses reported larger effect sizes when stimuli are presented for longer than 1000 ms,[21 23] whereas a recent analysis

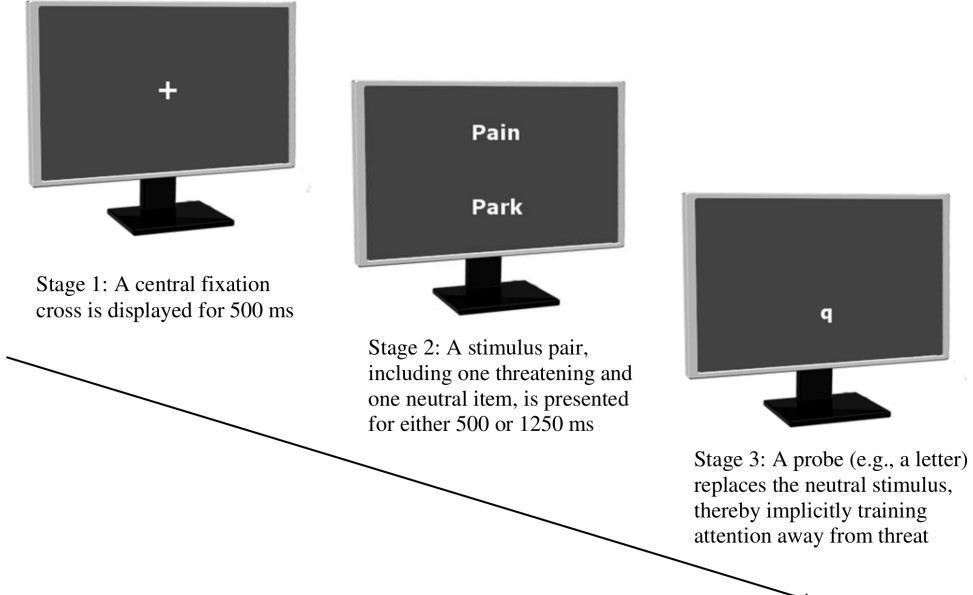

**Figure 1** A graphical representation of a typical visual-probe task training trial with linguistic stimuli.

found evidence of bias for presentation times of 500–1000 ms but not when stimuli are presented for longer than 1000 ms.[22] Despite this inconsistency, attentional biases are commonly shown in chronic pain, and preliminary research has explored the effects redirecting attention may have on pain and pain-related distress using attentional bias modification (ABM) techniques. ABM is a computer-based intervention which implicitly trains attention away from threat-related cues towards neutral information, and is frequently achieved through a modified version of the visual-probe task[24] (figure 1).

Two small randomised controlled trials (RCTs)[25 26] and one pilot study[27] have used the modified probe-classification version of the visual-probe task to assess ABM in adults with chronic pain. Carleton and colleagues[25] found participants receiving ABM showed significant reductions from pre-ABM to post-ABM in anxiety sensitivity and fear of pain, and marginally significant reduction in current pain severity. No significant reductions were found in illness/injury sensitivity or pain anxiety. Significantly more patients receiving ABM reported clinically significant change in current pain compared with those in the placebo group (44% vs 17%, respectively). Attentional bias scores were not recorded, and therefore the researchers could not comment on mechanisms of action. Sharpe and colleagues[26] found participants receiving ABM, compared with those in a placebo group, showed significant improvements in pain-related disability after the ABM. No significant improvements were found for pain, fear of pain, fear of reinjury, depression, anxiety, stress, anxiety sensitivity or pain self-efficacy. At 6-month follow-up (and after all participants had undergone eight sessions of cognitive behavioural therapy (CBT)) the ABM group, compared with the placebo group, showed significant improvements in disability and anxiety sensitivity, and marginally significant improvements in fear of

reinjury. No other significant effects were found, and the mechanism of action could not be established as ABM was not found to change patterns of attentional bias across time. Schoth and colleagues[27] reported statistically and clinically significant change from pre-ABM to post-ABM in pain intensity, pain interference, anxiety and depression. Assessed via the McGill Pain Questionnaire,[28] significant changes from pre-ABM to post-ABM were shown for total pain, affective pain and miscellaneous pain, but not sensory pain or evaluative pain. There was also no evidence that attentional biases changed across time.

One small RCT in adolescents with chronic pain randomised participants to either ABM, placebo training or waiting-list groups. No significant differences were found between the three groups in terms of pain, disability, pain catastrophising, anxiety, depression or objective measures of physical functioning (sit-to-stand task and cardio wall tasks), and no evidence was found that ABM changed attentional bias or attentional control compared with placebo and waiting-list conditions.[29] Overall, while some studies have provided support for the benefits of ABM in adults, the evidence in regard to specific outcome variables is mixed and the mechanisms of action have yet to be established. Given the methodological shortcomings and small sample sizes of previous studies however, further research is needed to determine the therapeutic benefits of ABM for patients with chronic pain, along with the optimal form ABM should take and the impact of age on outcomes and intervention engagement. Considering the optimal form of ABM, a review of the broader cognitive bias modification (CBM) literature identified 12 published meta-analyses, 11 examining ABM specifically.[30] Three out of six analyses found the number of sessions to be a significant moderator of CBM, and three out of nine analyses found the number of sessions to be a significant moderator of symptom outcomes.

Although evidence is therefore mixed, it is nevertheless worth exploring whether the number of sessions impacts the patterns of attentional bias and pain-related outcomes in individuals with chronic pain.

This article presents a study protocol for a randomised, double-blind, internet-delivered ABM intervention. The results of a pilot think-aloud study informing the design of the intervention are also presented. The aim of this RCT is to assess the efficacy of internet-delivered ABM in improving pain and pain-related distress in people aged 16–60 with CMSK, and will report a subgroup by treatment effects using the two age brackets of 16–24 and 25–60. While only limited evidence has been found for age as a moderator of CBM on bias change and symptoms,[30] it has been argued that the age of adolescence should be extended from 19 to 24 as the latter corresponds more closely to adolescent growth and popular understandings of this life phase.[31] Furthermore, although use of the internet has been growing among all age groups, there are nevertheless still some differences in internet use and activity according to age (eg, in the UK, 1% of those aged 16–24 do not use the internet, compared with 7% of those aged 45–54),[32] thus warranting a separate consideration of adolescent and adult participants. Overall, a diverse sample in terms of age, socioeconomic status and geographical region will be recruited, thus increasing generalisability of results to the CMSK population. The primary outcome measure is pain interference. Secondary outcome measures are pain intensity, state and trait anxiety, depression, pain-related fear and sleep impairment, and engagement to and satisfaction with the online visual-probe training task. Dose effects will be explored via the inclusion of 10-session vs 18-session conditions. It is hypothesised that participants in the intervention group, relative to those in the placebo group, will report significant reductions in pain interference, pain intensity, anxiety, depression, sleep impairment and pain-related fear at study endline. It is also hypothesised that dose will significantly moderate these findings, with greater effects in 18 compared with 10 sessions.

## METHODS AND ANALYSIS
### Participants
Participants will be recruited via press announcements, social media and patient organisations throughout the UK. Inclusion criteria will be (1) aged between 16 and 60 years old; (2) experiencing any type of CMSK (ie, any condition that involves pain lasting for more than 3 months and arises from the bones, muscles and/or joints); (3) able to sit at a personal computer for 40 min; (4) normal or corrected to normal vision; (5) access to the internet at least twice a week; (6) access to, and familiarity using, a Windows-based computer; (7) successful completion of primary school; and (8) living in the UK. Exclusion criteria will be (1) experiencing malignant CMSK (ie, pain due to a tumour); (2) a diagnosis of other forms of comorbid chronic pain (eg, chronic headache);

and (3) a diagnosis of any psychiatric disorder, either currently or within the last 5 years. There are no restrictions placed on concomitant care, and participants are not required to make any changes to current treatments they may be receiving.

### Patient and public involvement
Participant burden of the intervention and outcome measures were assessed using individual interviews and informal feedback from patients participating in two pilot studies.[27 33] Patients will not be involved in recruitment of participants or conduct of the study. Results of this study will be disseminated to participants through presentation at client and community forums.

### Study design and setting
The study is a double-blind RCT with four arms exploring the efficacy of online ABM training versus a placebo training condition theorised to offer no specific therapeutic benefits. The choice of comparator was selected as placebo training closely mimics ABM training, and is frequently used as a comparator in ABM studies (eg, refs 25 26 34 35). The study will take place in a location of the participant's choice (which we anticipate will typically be their home) and will be accessed via personal computer connected to the internet.

The study design has a 2×2×3 experimental structure, with multiple non-commensurate dependent variables measuring pain intensity and interference, emotional functioning, and pain-related disability. Treatment is a two-level factor (ABM, placebo), and the number of sessions is a two-level factor (10 sessions, 18 sessions), with measures recorded at three points in time (baseline, endline and 6-month follow-up). Measures in the 18-session groups (18AMG and 18PTG) will also be taken after session 10.

### Randomisation and sample size
Block randomisation of size four, performed via a randomisation command in the LifeGuide platform,[36] will be used to sequentially randomise eligible consecutive consenting participants to one of four groups: (1) 10-session attentional modification group (10AMG); (2) 10-session placebo training group (10PTG); (3) 18-session attentional modification group (18AMG); or (4) 18-session placebo training group (18PTG). This blocked randomisation will be performed separately for those aged 16–24 and those aged 25–60. An allocation concealment mechanism will be used to ensure neither participants nor researchers know the study group to which the next participant will be assigned. This will be automatically performed within the LifeGuide platform (see The intervention section), with participants provided access to appropriate AMG or PTG sessions. Participants randomised to either 10-session condition will receive ten 35 min online sessions across a 4-week period, while those randomised to either 18-session condition will receive eighteen 35 min online sessions across an 8-week period.

All online sessions will be administered twice a week on a separate set of days (Monday–Thursday, Tuesday–Friday). Participants randomised to either AMG condition will receive attentional training via the modified training version of the probe-classification visual-probe task, while those randomised to either PTG condition will receive placebo training via the standard version of the probe-classification visual-probe task. To maintain the overall quality and legitimacy of the intervention, code breaks (ie, unblinding) will only occur in exceptional circumstances. Adherence will be monitored electronically via LifeGuide.

Our previous lab-based, proof-of-concept ABM intervention found reductions in pain intensity and pain interference from pre-ABM to post-ABM were associated with large effect sizes,[27] although differences in methodology including delivery (online vs lab-based) and number of sessions warrant a more conservative estimate. On this basis, the study will recruit a minimum of 100 with 1:1:1:1 randomisation. This study is powered primarily to establish treatment efficacy for the primary outcome variable of pain interference by considering baseline and measures after session 10 between those randomised to an attentional modification arm and those randomised to a placebo arm. Randomisation should ensure baseline comparability on measures between these two groups, and a difference in beneficial outcomes between these groups will be modelled using a linear mixed model. For an assumed correlation of 0.7 between baseline and end of session 10 measures, a total of 40 per group (ie, 20 per randomised arm) would be needed to detect small to moderate standardised intervention differences (Cohen's $d$=0.25) with at least 80% power (alpha=0.05, two-sided). The recruitment strategy includes a 20% oversampling to mitigate a loss of power from loss to follow-up.

## Procedure
Interested individuals will contact the researchers on the details provided in study advertisements and will be assessed against the inclusion and exclusion criteria via telephone interview. If eligible, they will receive a copy of the participant information sheet detailing the requirements and procedure of the study, along with a link to the study website which provides further details on ABM and the research team. If the individual consents to take part, they will be required to create an online ABM account. During registration, participants will select their preferred days for training, which will be either (1) Monday/Thursday or (2) Tuesday/Friday.

Session 1 is the baseline session, and includes the first assessment visual-probe task measuring attentional biases prior to beginning the intervention, along with the questionnaire battery assessing the primary and secondary outcome measures. Randomisation will take place following session 1, where participants will be informed of the length of their training (ie, 4 or 8 weeks). Participants randomised to either of the 10-session conditions will complete four pictorial and four linguistic training sessions, while those randomised to either of the 18-session conditions will complete eight pictorial and eight linguistic training sessions (baseline and endline assessment sessions comprise the remainder of the 10/18 sessions). Participants' attentional biases will be reassessed at endline (ie, during the final session 10/18) using the same assessment version of the visual-probe task (ie, featuring the same stimuli) used at baseline and the same questionnaire battery. Participants randomised to either of the 18-session conditions will also have their attentional biases reassessed in session 10 after their training session.

Participants' satisfaction with the online training will also be assessed. Six months after the last session, participants will be invited to again complete the online assessment visual-probe task (same as baseline and final session 10/18) and questionnaire battery. Within 2 weeks of the completion of the 6-month follow-up session, participants will be invited to take part in a brief telephone semistructured interview including 15 questions spread across five thematic sections: (1) motivation and expectations about the study; (2) experience of the online study: person, environmental and lifestyle parameters; (3) experience of the visual-probe task; (4) understanding of the process of change; and (5) the visual-probe task as a potential therapeutic tool in pain management. The study flow is presented in table 1.

To retain as many participants as possible, reminder emails will be sent to participants 24 hours before each scheduled session becomes available, and also at 13:00 on the scheduled session date should the participant not have already logged into their internet-delivered ABMT (iABMT) account. An additional reminder email will be sent 1 week before their 6-month follow-up session is ready. Participants will also be sent encouragement emails informing them when they have reached specific milestones (ie, successfully completing 5 sessions for those randomised to complete 10 sessions; successfully completing 6 and 12 sessions for those randomised to 18 sessions). Participants will also have the option of receiving an SMS (short message service) on their mobile phone at 09:00 on the morning of each scheduled session.

## The intervention
The intervention will be hosted using LifeGuide and iSurvey platforms, both developed by the University of Southampton. LifeGuide is an open-source software which allows researchers to create online interventions.[36] iSurvey is a survey generation tool which allows researchers to create and disseminate questionnaires online. Participants will register for the intervention using LifeGuide, which will also host the majority of the questionnaires. iSurvey will be used to host the visual-probe tasks, along with the Engagement with the Online Training questionnaire and the Brief Pain Inventory,[37] which appear at the end of every visual-probe task session. The flow of sessions is provided in figure 2, and examples of web pages are shown in figure 3. On loading the intervention home

**Table 1** Schedule of enrolment, intervention and assessments

| Timepoint | Study period | | | | | |
|---|---|---|---|---|---|---|
| | Enrolment | Postallocation | | | | |
| | $-t_1$ | $t_1$ Baseline | $t_2$ Training | $t_3$ Evaluation | $t_4$ Follow-up | $t_5$ Study exit |
| **Enrolment** | | | | | | |
| Eligibility screen | X | | | | | |
| Informed consent | X | | | | | |
| Allocation | | X | | | | |
| **Interventions** | | | | | | |
| Attentional bias modification training | | | X | | | |
| **Assessments** | | | | | | |
| Demographic questionnaire | | X | | | | |
| EOT | | X | X | X | X | |
| BPI-SF | | X | | X | X | |
| HADS | | X | | X | X | |
| STAI | | X | | X | X | |
| MOS-SS | | X | | X | X | |
| FOP-III | | X | | X | X | |
| SOT | | | | X | | |
| Interview | | | | | | X |

$t_1$: baseline phase including standard assessment visual-probe task and questionnaires. Also includes randomisation to groups.

$t_2$: training phase including training visual-probe tasks.

$t_3$: evaluation phase including standard assessment visual-probe task and questionnaires. Participants randomised to 18-session conditions also complete evaluation phase during session 10.

$t_4$: 6-month follow-up including standard assessment visual-probe task and questionnaires.

$t_5$: telephone interview within 2 weeks of $t_4$.

BPI-SF, Brief Pain Inventory-Short Form; EOT, Engagement with the Online Training questionnaire; FOP-III, Fear of Pain Questionnaire III; HADS, Hospital Anxiety and Depression Scale; MOS-SS, Medical Outcomes Study-Sleep Scale; SOT, Satisfaction with the Online Treatment questionnaire; STAI, State-Trait Anxiety Inventory.

page, the individual is provided with a brief welcome to the website and rationale for the intervention. A recent review of cognitive bias meta-analyses concluded that effect sizes were smaller when biases were modified remotely compared with laboratory-based training.[30] We therefore decided to provide information on the rationale behind ABMT in order to increase motivation and reduce attrition as much as possible, which also mirrors clinical practice that involves educating patients into chronic pain and the psychological interventions used in its management. Individuals are able to either create an iABMT account or log into their existing account. Before any session participants are asked to indicate whether they have enough time and are ready to begin the session. Participants are only permitted to complete one session per day. Should a participant complete their missed session on a subsequently scheduled date (eg, completing their scheduled session from Monday on Thursday instead), all remaining sessions are shifted back accordingly. This process retains the structure of the intervention.

Guidance has been informed by research showing attention to be a limited capacity resource,[38 39] sleepiness to negatively influence concentration and performance,[40] alcohol to impair speed of information processing and cognition,[41] energy drinks to be associated with restlessness and nervousness,[42] caffeine to improve alertness,[43] large meals to be associated with impairments in cognitive functioning,[44] and hunger to influence patterns of attention.[45 46] Participants are provided with guidance to ensure concentration and comfort are maintained throughout each session: (1) allowing enough time for the session with no anticipated distractions; (2) adjusting screen brightness to match the brightness of the environment; (3) closing distracting applications such as Twitter and Facebook; (4) avoiding completing the session when tired; (5) not consuming alcohol at least 3 hours prior to the session; (6) caution in the use of energy drinks which may cause restlessness; (7) if the participant is a coffee or tea drinker, suggesting a cup 30 min prior to the session; and (8) avoiding completing the session when hungry or directly after a meal.

### Visual-probe tasks

The flow of visual-probe sessions is illustrated in figure 2. In total there will be eight linguistic and eight pictorial probe-classification visual-probe training tasks used in

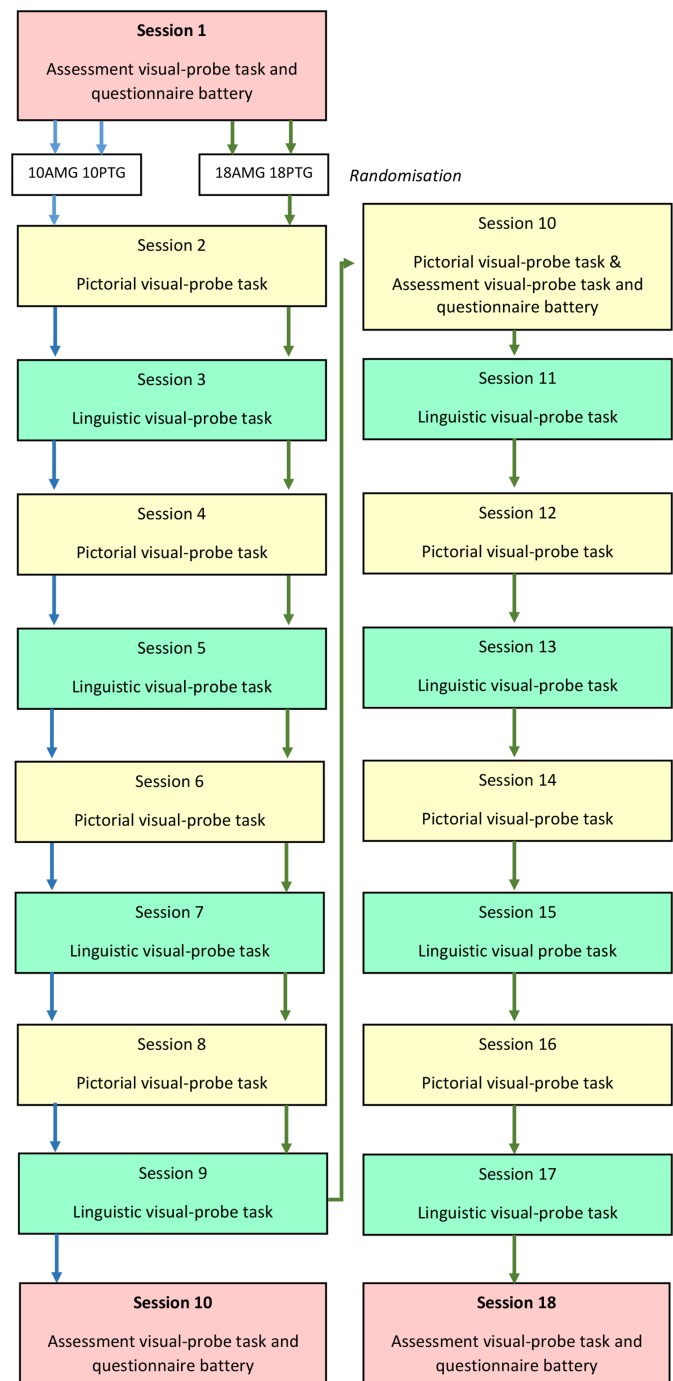

**Figure 2** Flow of the visual-probe task sessions for the 10-week and 18-week groups in the study. AMG, attentional modification group; PTG, placebo training group.

the study, each with different stimuli. A complete description of the stimuli including ratings of valence, arousal, pain intensity, written frequency and low-level features is provided in online supplementary material 1. To briefly summarise, each linguistic visual-probe task includes 32 threatening/neutral and 16 neutral/neutral word pairs. Each task includes eight threatening words from each of the following categories: sensory pain, affective pain, health threat and general threat. Each pictorial task includes 32 threatening/neutral and 16 neutral/

neutral image pairs. Each task includes eight threatening images from the following categories: musculoskeletal pain, facial expressions of pain, health threat and general threat. A total of 384 trials are included in each task, with each stimuli pair presented four times for 500 ms and four times for 1250 ms. Within each exposure duration, each stimulus appears twice in the upper location of the screen and twice in the lower location.

A separate linguistic visual-probe task will be used in the three assessment sessions, featuring different stimuli to the linguistic training visual-probe tasks. The use of separate assessment and training stimuli is necessary to establish whether training effects generalise to novel stimuli not used in the actual training. The same assessment stimuli will be used in all assessment sessions. The ABM training version of the visual-probe task to be completed by AMG groups includes three main stages per trial (figure 1). Following a 500 ms initial fixation point, a stimulus pair is presented in distinct locations (eg, above and below the initial fixation point) for either 500 or 1250 ms. The stimulus pair may be either words or images, although critically includes one threat-related and one neutral stimulus. Stimuli presentation times of 500 and 1250 ms are used as these are the most common stimuli presentation times adopted in chronic pain visual-probe studies (eg, refs [18 20 47]) and will allow us to closely compare our results with those of previous research. Immediately following their presentation, the stimuli disappear and a probe appears (either 'p' or 'q') in the location of the neutral stimulus. Participants use corresponding keys on their keyboard to indicate the probe letter as quickly and as accurately as possible. In the standard version of this paradigm completed by PTG groups and used in the assessment sessions, the probe replaces the neutral and threat-related stimulus an equal number of times. The probe-classification version of the visual-prove task is argued to encourage a more even monitoring of the display than the probe-position version (ie, which requires participants to indicate the location of a dot, either left/right or up/down), as for the latter participants can adopt biased monitoring strategies favouring one particular region over the other.[48] As any such biases in monitoring may affect training, we will use the probe-classification version of the visual-probe task.

## Outcome measures
The primary outcome measure of pain interference will be assessed by the Brief Pain Inventory-Short Form.[37] Pain interference will be scored as the mean of seven interference items (ie, general activity, walking, work, mood, enjoyment of life, relations with others and sleep). Following guidance in the Brief Pain Inventory User Guide[49] and also Initiative on Methods, Measurement, and Pain Assessment in Clinical Trials (IMMPACT) recommendations for assessing pain in clinical trials,[50] we will include individual assessments of 'worst', 'least', 'average' and 'now' (ie, current) pain intensity as secondary outcome measures. Additional secondary outcome measures of

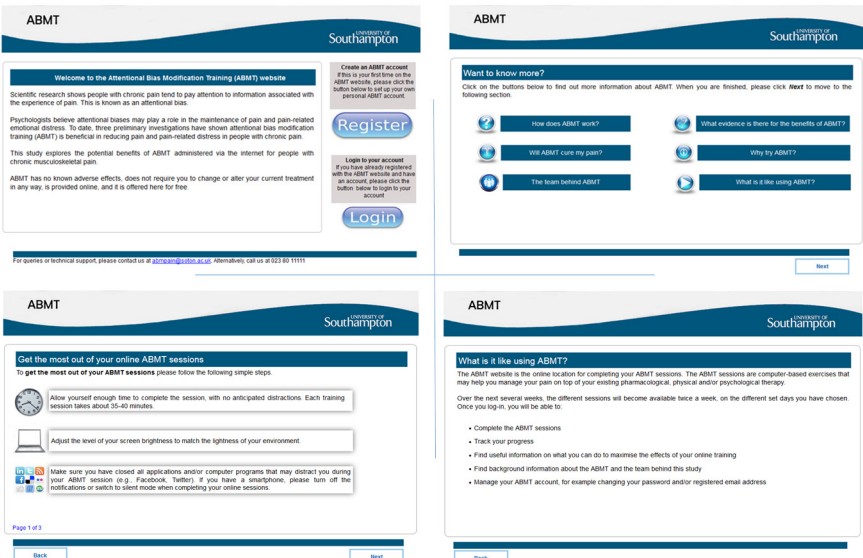

**Figure 3** Examples of LifeGuide web pages included in the internet-delivered attentional bias modification training intervention.

state and trait anxiety will be assessed by the State Trait Anxiety Inventory,[51] depression by the Hospital Anxiety and Depression Scale,[52] pain-related fear by the Fear of Pain Questionnaire III,[53] sleeping impairment by the Medical Outcomes Study-Sleep Scale,[54] and engagement to and satisfaction with the online visual-probe training task by the Satisfaction with the Online Training questionnaire (SOT)[55] and the Engagement with the Online Training questionnaire (EOT). A demographic questionnaire developed by the researchers will be used at baseline to collect information on participants' age, gender, ethnicity, socioeconomic status, place of residence and pain characteristics. Aside from the SOT and EOT, questionnaires in the 10-session groups (10AMG and 10PTG) will be administered to participants at baseline, after 10 sessions and at 6-month follow-up. In the 18-session groups (18AMG and 18PTG), questionnaires will be administered to participants at baseline, after 10 sessions, after 18 sessions and at 6-month follow-up. At endline for each participant, we will also assess whether the participant and researchers became aware at any point prior to endline which group they had been allocated to. Full details on each outcome measure are provided in online supplementary material 2.

### Think-aloud study

A qualitative think-aloud study[33] including a short semi-structured interview was conducted following the development of the initial version of the intervention with the aim to understand participants' first impressions and attitudes towards the intervention, and to collate feedback that could be used to make improvements. Details and outcomes of this study are provided in online supplementary material 3.

### Analyses of the RCT data
#### Quantitative analyses

A full and detailed statistical analysis plan will be written prior to trial closure. In brief, the data analysis will be performed on an intention-to-treat (ITT) basis to provide an unbiased estimate of the treatment effect.[56] Per-protocol analyses will also be performed (comprising those correctly randomised, without protocol violations, and providing complete data on the primary outcome measure). Sweetman and Doig[57] have identified five types of protocol violations, namely violations in (1) enrolment (ie, failure to correctly apply eligibility criteria resulting in the enrolment of an inappropriate participant); (2) randomisation (ie, violation of intended randomisation sequence); (3) study intervention (ie, dosing, timing or delivery errors attributable to members of the research team); (4) patient compliance (ie, participants failing to comply with the trial protocol or other requirements of participation in the trial, such as skipping scheduled appointments or sessions); and (5) data collection (ie, a failure by the research team to comply with prespecific trial guidelines for data collection and/or outcome evaluation due to avoidable reasons). Any such violations will be clearly reported.

Analyses undertaken will include the following:
1. The primary analysis on pain interference will use the initial 2×2×2 structure comprising stage (baseline, 10 weeks), randomised treatment (attentional bias modification training (ABMT), placebo) and randomised number of sessions (10, 18). This comparison will be extended to compare 10AMG and 10PTG after 10 sessions, and to compare 18AMG and 18PTG after 10 sessions.
2. An analysis of pain interference will use the 2×2×3 structure comprising randomised treatment (ABMT,

placebo), randomised number of sessions (10, 18) and stage (baseline, endline, follow-up).

3. An assessment of change in pain interference, both within and between groups, for 18AMG and 18PTG, between the end of session 10 and endline.

4. The four randomised arms compared against one another on pain interference at each of baseline, after session 10, endline and follow-up.

5. An age (16–24, 25–60) subgroup analysis using an age subgroup by treatment effect interaction. These analyses will be an extension of (1), (2) and (3) to incorporate age as a subgroup factor.

The same analyses would apply to the secondary outcome measures. The data will be represented descriptively and graphically by arm, and by treatment and age, and standardised effect sizes reported.

Linear mixed model with random intercepts will be used to analyse the data with, as appropriate, treatment (ABMT, placebo), number of sessions (10, 18) and assessment stage as fixed factors. This approach avoids listwise deletion which would apply in a repeated measures analysis of variance approach. The linear mixed modelling approach uses all of the available data and adheres to ITT. As a secondary analysis we will used multiple imputation chained equations to impute outcome data[58 59] and use these imputed data sets in the linear mixed model. Pairwise comparisons will follow the same procedure. An assessment of the sensitivity of findings to missing data mechanisms will be undertaken.[60 61] Specifically, we will use the strategy discussed by Morris and colleagues[61] whereby multiply imputed values are altered by a randomised group-dependent value prior to analysis. We will run this sensitivity analysis over a factorial arrangement of group-dependent values constraining any altered imputed values to be in the valid response range. The instance of the group-dependent values both being equal to zero will correspond to multiple imputation under missing at random (MAR), and non-zero values would reflect deviations away from this assumption. The analysis plan would apply to the primary outcome measure and secondary outcome measures.

Similar to previous studies,[25 27] the Reliable Change Index[62] will be used to assess primary and secondary outcome measures for clinically significant changes. Here, a clinically significant result is a post-treatment score falling outside 2 SD of the mean population of interest (in this study, 2 SD outside pretreatment means). In contrast to tests of statistical significance comparing group means, tests of clinical significance explore effects of treatment on the individual.[63]

For visual-probe data, and as per previous research,[19 27] practice and incorrect trials will be removed prior to analyses. Box and whisker plots for overall data will be used to reveal overall outliers, which will be removed. Following this, mean response times will be computed for each participant, with any response >3 SD from their individual mean also removed as outliers. This process ensures extremely quick or slow responses do not unduly bias

the results, which are typically removed when cleaning and screening visual-probe data in pain-related research (eg, refs [18 64–67]). An attentional bias index will then be computed for each stimulus and presentation time condition using the following equation: $(TuPl-TlPl)+(TlPu-TuPu)/2$. Here T is the threatening stimulus, P is the probe, u is the upper position and l is the lower position. The attention capturing quality of threatening stimuli is measured by subtracting the mean probe classification time for congruent trials from the mean probe classification time of incongruent trials.[21]

Split-half reliability will be computed via a bootstrap procedure, by randomly splitting the total number of trials in two halves, such that each half has the same number of congruent and incongruent trials. Attentional bias scores will be computed for each half and the correlation between them calculated across participants. This procedure will be repeated 100 times, and an average split-half correlation computed.

### Qualitative analyses
Qualitative analysis will be used to detail participants' experiences completing the iABMT intervention. Specifically, thematic analysis using inductive coding will be used to identify themes present in the data obtained from the semi-structured interviews and the two open-ended questions included in the SOT questionnaire. Guidelines provided by Braun and Clarke[68] will be followed. Following verbatim transcription of all interviews, initial codes will be generated. Potential themes and subthemes will then be identified from the initial coding, which subsequently will be reviewed and refined. The validity of the themes in relation to the data set will also be considered. Each theme and subtheme will then be named and defined, and a thematic map of the data finalised. Examples from the data will be illustrated for each theme and subtheme in the final report.

### Ethics and dissemination
The study protocol was registered with ClinicalTrials.gov, with a planned start date of September 2020 and a planned end date of September 2022. All data will remain confidential and stored on password-protected systems and databases. LifeGuide and iSurvey servers both use HTTPS (hypertext transfer protocol secure) connections for security purposes. All data will be anonymised prior to dissemination, and personally identifiable information not known to anyone other than the researchers. Participants may withdraw from the intervention at any point by contacting the researchers via email or telephone, or by simply opting not to log into their iABMT account for their scheduled sessions. Participants are also encouraged to contact the researchers should they experience any distress or discomfort completing the intervention. As per university policy, any adverse events, harms or complaints arising from participation in the intervention will be reported to the University of Southampton Research Governance Office. Findings will be published in the most relevant, high-impact, peer-reviewed journals and presented at relevant conferences.

Lay reports will be written for interested pain charities and patient support groups.

**Contributors** All authors have contributed to the development of the study protocol. CL, TG and DES conceived the project. CL and DES registered the trial and are responsible for recruitment and implementation of the protocol. TG was primarily responsible for the development of the online materials, including development and validation of linguistic and pictorial stimuli used in the visual-probe tasks. TG also assisted in programming the LifeGuide components. JZ was responsible for programming the visual-probe tasks to be used in the intervention and assisted in programming the LifeGuide components. FH conducted the think-aloud study. PW was responsible for the development of the statistical analysis plan. All authors contributed to the writing of this manuscript and approve its final form. These same authors will be included in future publications presenting the intervention results and will have access to the cleaned study data sets.

**Funding** This study was funded by the Economic and Social Research Council.

**Competing interests** None declared.

**Patient consent for publication** Not required.

**Ethics approval** This study has been approved by the University of Southampton Research Ethics Committee (ERGO ID: 26486). Ethical approval will be sought before any protocol modifications are made, and the study protocol will be updated at ClinicalTrials.gov. Participants will provide informed consent prior to taking part in the study.

**Provenance and peer review** Not commissioned; externally peer reviewed.

**ORCID iD**
Christina Liossi http://orcid.org/0000-0003-0627-6377

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
