## [Reviewer comments · BMJ Open]

ARTICLE DETAILS

TITLE (PROVISIONAL)	Internet-delivered attentional bias modification training (iABMT) for the management of chronic musculoskeletal pain: a protocol for a randomised controlled trial
AUTHORS	Liossi, Christina; Georgallis, Tsampikos; Zhang, Jin; Hamilton, Fiona; White, Paul; Schoth, Daniel Eric

VERSION 1 – REVIEW

REVIEWER	Lauren Heathcote Stanford University School of Medicine, USA
REVIEW RETURNED	06-Apr-2019

GENERAL COMMENTS	The authors present a protocol for an RCT of internet-delivered ABM training for individuals with chronic pain. The protocol is well-written and closely follows the preregistered study aims on clinicaltrials.gov. The hypotheses and statistical plan are clear. I have just a few small comments to improve clarity on a few issues: 1) Description of the attention control arm. I am a little confused that the placebo training arm is referred to as an 'attention control' arm. Typically, an attention control arm in an RCT is named as such to match the attention that patients receive from clinicians in face-to-face treatment. I understand that it can also refer to the attention received from researchers, therefore representing the effect of being engaged in a clinical trial - I assume this is how the authors intend to use this wording for this study? But it is clear that participants in this arm are receiving a placebo version of the training paradigm, meant to control for many other factors (e.g., repeatedly viewing the stimuli). So the term 'attention control arm' seems inappropriate. The slight confusion is also that 'attention control training' is another active type of cognitive training that is common in this field, referring to the training of effortful attention control. Another confusion is that the authors sometimes refer to the attention control arm as a placebo training arm (and the latter is how it is referred to in clinicaltrials.gov) and other times as the 'standard version of the paradigm'. Can the authors clarify their choice and descriptions? 2) Description of the allocation concealment. The authors simply write that 'an allocation concealment mechanism will be used' to ensure that the trial is double-blind, but it is unclear what this mechanism entails. I assume that some members of the research team will be unblinded, so that they can send participants the correct version of the task? Or will this be automated in some way? In some cases the researchers who conduct analyses are blinded while researchers administering (i.e., sending) the tasks are not - is this the case here?
---

	3) Intention-to-treat analyses. I found it confusing to understand how the authors plan to conduct intention-to-treat analyses. When describing the overall ANOVAs, they report that only data from completers will be included. The intention-to-treat analysis is mentioned after presenting the post-hoc analyses. Will the authors conduct intention-to-treat analyses for the primary ANOVA analyses? If so, do they plan to use the BOCF (baseline observation carried forward) or LOCF (last observation carried forward) method? 4) In my opinion, the authors present a slightly unbalanced view of the current evidence for ABM training. They refer largely to the observed positive effects of ABM training in chronic pain samples so far (apart from the one pediatric study). Yet, the evidence is quite mixed in that effects are not consistently observed on primary outcomes, and that the supposed mechanism of effect (tested by mediation analyses) is typically not supported. Can the authors include a more balanced summary of the evidence base, including the outcomes that have not shown an effect through ABM, and the issue of mechanism? 5) The authors do not seem to intend on performing mediation analyses to examine potential mechanism of effect. They could, for example, examine whether improvements in ABM training are mediated by a change in bias (per the assessment sessions). Can the authors justify why this is not part of their study, especially given the issues described in comment 4?
--	---

REVIEWER	Louise Sharpe The University of Sydney, Australia
REVIEW RETURNED	21-Apr-2019

GENERAL COMMENTS	The authors propose a randomized controlled trial of attention bias modification for patients with musculoskeletal pain. They plan to recruit 160 participants (80 in the age range 16-24 and 80 between the ages of 24 and 60) and allocate them to one of four conditions: Two ABM conditions (one with 10 training sessions and 1 with 18 training sessions) and two placebo conditions (an unbiased dot-probe task with either 10 or 18 training sessions). Therefore, although some of the interactions (e.g. dosage x group x age) may be underpowered, there should be sufficient power to answer the primary question of whether ABM is efficacious for people with chronic musculoskeletal pain compared to placebo, when administered via the internet, assuming at least a medium size effect. To date, there have been a number of laboratory studies with promising results, but there have only been a handful of small randomized trials of ABM in people with chronic pain. Hence, there is a great need for a large trial of ABM with chronic pain patients. As such, the proposed trial will be a welcome addition to the literature. However, I feel that there could be more information about the specific nature of the training protocol, and I think that the authors could benefit from looking at the much more extensive literature in the anxiety domain to support (or otherwise) some of the hypotheses. Major Issues:  1. The authors state that it is important for the impact of age on outcomes to be determined but provide no rationale for this. It is
---

true that meta-analyses confirm that the effect sizes for the presence of an attentional bias in people with anxiety compared to those without is moderated by age, but only in children. Similarly, meta-analyses demonstrate that ABM is not effective in children (Cristea, et al. (2015). *Journal of Child Psychology and Psychiatry*, 56(7), 723-734) in contrast to meta-analyses in adults [of which there are 13 according to a recent systematic review]. Since the only trial in adolescents in pain failed to find a therapeutic effect of ABM (Heathcote et al., 2018), it is possible that ABM is effective in adults but not children and adolescents. If this is the case, the most likely mechanism is attentional control – that is, that a certain level of attentional control is required in order for ABM to be effective. However, the age ranges in the current study are 16-24 vs older. Attentional control is reasonably well developed by age 16 and so it is unclear to me why one would expect an age effect for these age groups – and, if there was one, it would be important to ensure that duration of the chronic pain was not the driver of this effect (as age and duration could be associated). The rationale for this hypothesis and the relevant age ranges needs to be explained.

2. The authors hypothesize that dose will moderate the effect, but it is unclear on what basis they make this prediction, since the two individual meta-analyses that investigated dose found it unrelated to outcome (Heeren, A., Mogoșe, C., Philippot, P., & McNally, R. J. (2015). *Clinical psychology review*, 40, 76-90) and (Mogoșe, C., David, D., & Koster, E. H. (2014). *Journal of Clinical Psychology*, 70(12), 1133-1157.). Can the authors specify on the basis of what data they make this hypothesis? A rationale for 10 versus 18 sessions would also be helpful.

3. I am a bit confused by the term “attention control training” as the “placebo”. In the PTSD literature, the “placebo” training has been shown to be effective and as such, the ABM placebo has been renamed attention control training, to recognise the likely mechanism for this effect, training attentional control (Badura-Brack et al. (2015). *American Journal of Psychiatry*, 172(12), 1233-1241.), but I think it will be confusing to readers to call it “attention control training” when essentially this condition is being used as a placebo. I think that the authors should state that. I think that the protocol would make more sense to readers if the authors adopted the usually used acronyms of ABM (attention bias modification) and placebo training (PT).

4. I am confused by the power analysis. Why is the study powered on the 10 session versions and not the 18 sessions? And what is the basis for expecting a medium effect size? While the effect sizes vary in the literature and in the meta-analyses, in our recent systematic review, we found the range of effect sizes across meta-analyses from small (Cohen's $d = 0.16$) to small to moderate (Cohen's $d = 0.41$) for symptom reduction (as proposed here) [Jones & Sharpe (2017). *Journal of Affective Disorders*, 223, 175-183.]. These meta-analyses were not in pain, but the effects appear more robust in anxiety than other areas where they have been used – so on what basis do the authors expect such large effects? This question is particularly pertinent since delivery via the internet in other areas has been found consistently to result in smaller effects than delivery via other modes.

5. Will the authors assess blindness for both participants and researchers at the end of the study?

6. In ABM training studies, typically participants have not been given information about ABM given that the training is surreptitious and the mechanism is argued to be implicit. Can the authors state

	what is the information given to participants at baseline through the website and what is the rationale for this? Have the authors considered how this might affect the participants and the study outcomes? It may be that the lack of a rationale (and therefore motivation) to complete the task is one reason for the failure of internet-based protocols to be successful, so this might be helpful, but should be discussed. 7. The authors state that they will use the “same” task at pre- and post-treatment, but it is unclear whether they mean the same stimuli. Typically, the stimuli are changed in the post-treatment assessment phase from both the pre-treatment assessment phase and training phase. Can the authors please clarify? 8. I could not find sufficient details in the protocol about the actual ABM task used. As I understand it, the ABM will have 8 pairs of pictures and 8 pairs of words (16 unique combinations). For each combination, if they are presented in the four necessary combinations for the placebo, there will be 64 trials. I do not see how the ABM could take 25 minutes. Can the authors state more clearly, the number of trials, and exact parameters of ABM training in order to allow other authors to replicate. I was also confused as to whether there were 18 different combinations of the 16 pairs, or whether the training simply repeated the same stimuli. Finally, the authors state that they are using threat-related stimuli and I commend them for all the included data that they provide in supplementary materials, but I still found it hard to be clear what proportion of stimuli were pain-related, and it was only upon reading the supplementary materials that I realised that there were neutral/neutral trials included. I think there just needs to be more clarity in the paper itself. 9. It would be excellent to see the authors state some pre-specified mediation analyses, relating to changes in biases – particularly if they have hypotheses relating to biases at 500msec vs 1250 msec. 10. I was a little unconvinced about the planned analyses. In terms of the changes in biases and any mediation analyses, I can see that a completers’ analysis makes sense, but in terms of efficacy, an intention to treat analysis would be required. I thought that the manuscript was unclear on this. Can the authors also state why they are not using a linear mixed models analysis? Minor Issues: 1. The authors state the results of the available meta-analyses, but omit the most recent meta-analysis (see Todd et al., (2018). Attentional bias to pain-related information: A meta-analysis of dot-probe studies. Health psychology review, 12(4), 419-436.). Further, they state that attentional biases are present “particularly at longer stimuli presentation times (i.e., > 1000 ms) associated with maintained attention as indicated in meta-analyses.^{21 22}” This is not entirely accurate since the larger two meta-analyses of Crombez et al. (2013) and Todd et al. (2018) do not find that biases are more evident at longer presentations, and actually much of the eye tracking data supports vigilance as the primary attentional bias, where results about difficulty disengaging is less clear. So I would omit this from the sentence. 2. Line 43-44 on page 4 ends “another in and pain intensity”. There seems to be a word missing. 3. The authors suggest “The stimulus pair may be either words or images, although critically includes one threat-related and one neutral stimulus. 500 and 1250 ms presentation times are used to explore biases in initial orienting of attention and maintained attention respectively.²¹” The authors themselves did suggest that
--	---

	these durations match biases in initial orienting and maintained attention, but most researchers would suggest that this was an oversimplification. It is a strength that the authors include both, but they should definitely rephrase this. I would recommend the authors read the seminal paper by Cisler and Koster, and adopt their language about the different nature of attention biases – or, since it is not entirely relevant to the protocol paper, perhaps they could omit it altogether. (Cisler, J. M., & Koster, E. H. (2010). Clinical psychology review, 30(2), 203-216.) 4. Will adverse events be recorded?
--	--

REVIEWER	Dimitri Van Ryckeghem Maastricht University
REVIEW RETURNED	01-May-2019

GENERAL COMMENTS	I would like to thank the editor for giving me the opportunity to review this protocol for internetdelivered attention bias modifican in chronic pain patients. The protocol is well developed, however lacks sometimes a clear rationale for particular decisions made. Below I describe the parts for which I think an additional rationale would be helpful for the readers. 1. The authors describe this study as the first study investigating internet-delivered ABM in chronic pain patients. Although I agree that systematic research on ABM in chronic pain is lacking, I would like to draw the attention of the researchers to following studies. First, a study of Heathcote and colleagues (2018) provides findings of ABM in adolescents suffering chronic pain. Second, there is also an ongoing/finalized internet delivered ABM trial by Asmundson and colleagues (see Van Ryckeghem et al., 2018). Therefore, I would suggest authors do not present this study as the first 2. The authors mention that the inclusion of a diverse chronic pain sample is strength of current study? It would be helpful to have a rationale for this suggestion. Indeed, having a very heterogeneous sample may also be a drawback for clear conclusions. Could authors better motivate their choice for a heterogeneous sample? 3. In the introduction, only the positive findings of previous ABM studies have been presented. I would suggest that authors also indicate the negative and null-findings of each of these studies. Indeed, many studies include a large number of variables of which only few are significant. Providing also an overview of these negative and null-findings results in a more complete picture of the findings of ABM research to-date. 4. I was surprised that the authors did not report a change in attention bias as an outcome of attention bias modification training. Why did authors not include this as an outcome? Do authors think they have sufficient power to detect a change in AB? In particular, since the correlation between AB over time is low (see Dear et al., 2011), Could authors discuss this further. 5. Could authors provide a rationale for the different age groups. Why did authors split groups based on age. Again, do authors have sufficient power to detect differences between groups? Or do they not expect a difference? What is the rationale for this particular groups? Similarly, could authors discuss the
--

	rationale for the dose? These doses are relatively large (see also doses given in anxiety and depression) compared to previous interventions. 6. In contrast to many of the previous studies, the authors have chosen for a presentation time of 500 and 1250 ms (mixed). Again, a rationale for this decision may be helpful. In particular, because based upon a different presentation time other processes may be targeted (ruminative vs early attention processes). Previous research has systematically opted for a presentation time of 500 ms. Authors may be interested in the findings of a recent review on Todd and colleagues (2018) discussing the impact of experimental procedures on the presence and magnitude of AB. 7. Authors focus on musculoskeletal pain. How will authors deal with comorbid pain (e.g., Headache, neuropathic pain,..). Will participants be excluded on this basis? Please provide clarification on the inclusion and exclusion criteria and the rationale for these criteria. 8. Related to this, how will a diagnosis of any psychiatric disorder (e.g., depression and anxiety disorders are very common) will be determined (self-report, MD, ..) furthermore, what is meant with psychiatric therapy. Please clarify.
--	--

REVIEWER	Lucy Busija Monash University, Australia
REVIEW RETURNED	10-May-2019

GENERAL COMMENTS	This is a clearly written protocol and the study is well-designed overall. Aims and hypotheses are clear. Sample size calculation, randomisation, blinding procedures, and statistical analyses are appropriate and are generally described in sufficient detail. However, some aspects of the study design are not clear and require further details. My specific quires are presented bellow. Recruitment: Not clear how interested participants will express their interest in taking part in the study Attrition: Please clarify the anticipated attrition rate for the study sample and how you will ensure that attrition does not compromise study power. While the use of multiple imputation to handle missing data is appropriate, multiple imputation is known to increase standard errors of estimates, with the resultant loss of power. What measures are in place to ensure that attrition does not compromise the study power at the end of follow up? Randomisation: The authors propose to analysed data for participants aged 16 – 24 and 25 – 60 separately. How will you ensure balanced allocation to the intervention and placebo groups in each age group? Data handling: Please clarify the process of outlier removal. If any 'unusual' observations are to be removed prior to analyses, what measures are in place to ensure that intent to treat principle is adhered to during data analysis (ie, does your process of data cleaning safeguard against the possibility of removing participants with 'unusual' response patterns from analyses, as opposed to only removing 'unusual' trials for some of the participants)? Data analysis: With the application of Bonferroni correction to post hoc tests, will your study have sufficient power to detect meaningful differences? Minor comments
---

	Throughout text: 'randomised control trial' should be 'randomised controlled trial'; please check the manuscript carefully and correct the terminology.
--	---

VERSION 1 – AUTHOR RESPONSE

Reviewer: 1. Reviewer Name: Lauren

Heathcote

- 1) *Description of the attention control arm. I am a little confused that the placebo training arm is referred to as an 'attention control' arm. Typically, an attention control arm in an RCT is named as such to match the attention that patients receive from clinicians in face-to-face treatment. I understand that it can also refer to the attention received from researchers, therefore representing the effect of being engaged in a clinical trial - I assume this is how the authors intend to use this wording for this study? But it is clear that participants in this arm are receiving a placebo version of the training paradigm, meant to control for many other factors (e.g., repeatedly viewing the stimuli). So the term 'attention control arm' seems inappropriate. The slight confusion is also that 'attention control training' is another active type of cognitive training that is common in this field, referring to the training of effortful attention control. Another confusion is that the authors sometimes refer to the attention control arm as a placebo training arm (and the latter is how it is referred to in clinicaltrials.gov) and other times as the 'standard version of the paradigm'. Can the authors clarify their choice and descriptions?*

We now refer to this as 'placebo training' and the 'placebo training group (PTG)' in all documents.

- 2) *Description of the allocation concealment. The authors simply write that 'an allocation concealment mechanism will be used' to ensure that the trial is double-blind, but it is unclear what this mechanism entails. I assume that some members of the research team will be unblinded, so that they can send participants the correct version of the task? Or will this be automated in some way? In some cases the researchers who conduct analyses are blinded while researchers administering (i.e., sending) the tasks are not - is this the case here?*

We have clarified that randomisation is automatically performed via the Lifeguide platform. After randomisation, participants are automatically provided access to the correct versions of the tasks.

- 3) *Intention-to-treat analyses. I found it confusing to understand how the authors plan to conduct intention-totreat analyses. When describing the overall ANOVAs, they report that only data from completes will be included. The intention-to-treat analysis is mentioned after presenting the post-hoc analyses. Will the authors conduct intention-to-treat analyses for the primary ANOVA analyses? If so, do they plan to use the BOCF (baseline observation carried forward) or LOCF (last observation carried forward) method?*

Based on the reviewers' comments we have now amended our analysis plan, and will use linear mixed model with random intercepts to analyse our data. We have described in more detail our intention-totreat analysis. As per current practice, a comprehensive Statistical Analysis Plan (SAP) will be written before the data analysis commences.

- 4) *In my opinion, the authors present a slightly unbalanced view of the current evidence for ABM training. They refer largely to the observed positive effects of ABM training in chronic pain samples so far (apart from the one pediatric study). Yet, the evidence is quite mixed in that*

effects are not consistently observed on primary outcomes, and that the supposed mechanism of effect (tested by mediation analyses) is typically not supported. Can the authors include a more balanced summary of the evidence base, including the outcomes that have not shown an effect through ABM, and the issue of mechanism?

Thank you. We now include a much more detailed description of the results of former ABM studies in the Introduction.

- 5) *The authors do not seem to intend on performing mediation analyses to examine potential mechanism of effect. They could, for example, examine whether improvements in ABM training are mediated by a change in bias (per the assessment sessions). Can the authors justify why this is not part of their study, especially given the issues described in comment 4?*

A comprehensive Statistical Analysis Plan (SAP) will be written before study closure and will be circulated for review by the Trial Steering Committee and Trial Management Group . Secondary analyses including mediation analysis will be considered at that stage. Our primary aim is to first establish whether our intervention has therapeutic benefits.

Reviewer: 2. Reviewer Name: Louise Sharpe

1. The authors state that it is important for the impact of age on outcomes to be determined but provide no rationale for this. It is true that meta-analyses confirm that the effect sizes for the presence of an attentional bias in people with anxiety compared to those without is moderated by age, but only in children. Similarly, meta-analyses demonstrate that ABM is not effective in children (Cristea, et al. (2015). *Journal of Child Psychology and Psychiatry*, 56(7), 723-734) in contrast to meta-analyses in adults [of which there are 13 according to a recent systematic review]. Since the only trial in adolescents in pain failed to find a therapeutic effect of ABM (Heathcote et al., 2018), it is possible that ABM is effective in adults but not children and adolescents. If this is the case, the most likely mechanism is attentional control – that is, that a certain level of attentional control is required in order for ABM to be effective. However, the age ranges in the current study are 16-24 vs older. Attentional control is reasonably well developed by age 16 and so it is unclear to me why one would expect an age effect for these age groups – and, if there was one, it would be important to ensure that duration of the chronic pain was not the driver of this effect (as age and duration could be associated). The rationale for this hypothesis and the relevant age ranges needs to be explained.

We agree in principle with your comments; however, given that there is only one study that has explored the efficacy of ABMT in paediatrics and that the anxiety literature is not directly relevant and transferable to our population, we have decided to explore these questions further. Chronic MSK conditions of older adolescents (i.e., 16-24) in addition to being of shorter duration tend to also be different in other aspects, including their pathophysiology and biopsychosocial formulation, and therefore are possibly more responsive to ABMT. Even if we find that the duration of chronic pain moderates the therapeutic benefit it would be useful to know and will push for early introduction of ABMT to paediatric chronic pain services for example.

2. The authors hypothesize that dose will moderate the effect, but it is unclear on what basis they make this prediction, since the two individual meta-analyses that investigated dose found it unrelated to outcome (Heeren, A., Mogoşşe, C., Philippot, P., & McNally, R. J. (2015). *Clinical psychology review*, 40, 76-90) and (Mogoşşe, C., David, D., & Koster, E. H. (2014). *Journal of Clinical Psychology*, 70(12), 1133-1157.). Can the authors specify on the basis of

what data they make this hypothesis? A rationale for 10 versus 18 sessions would also be helpful.

Given the limited available evidence on the efficacy of ABMT we believe the issue of a dose dependent effect is still an open question and we therefore decided to explore it. 10 vs 18 is exploratory and based on the average sessions for CBT 8 (+ 2 assessment sessions) and double the dose i.e., 16 (+ 2 assessment sessions).

3. *I am a bit confused by the term “attention control training” as the “placebo”. In the PTSD literature, the “placebo” training has been shown to be effective and as such, the ABM placebo has been re-named attention control training, to recognise the likely mechanism for this effect, training attentional control (Badura-Brack et al. (2015). American Journal of Psychiatry, 172(12), 1233-1241.), but I think it will be confusing to readers to call it “attention control training” when essentially this condition is being used as a placebo. I think that the authors should state that. I think that the protocol would make more sense to readers if the authors adopted the usually used acronyms of ABM (attention bias modification) and placebo training (PT).*

We now refer to this as ‘placebo training’ and the ‘placebo training group (PTG)’ in all documents.

4. I am confused by the power analysis. Why is the study powered on the 10 session versions and not the 18 sessions? And what is the basis for expecting a medium effect size? While the effect sizes vary in the literature and in the meta-analyses, in our recent systematic review, we found the range of effect sizes across metaanalyses from small (Cohen’s $d = 0.16$) to small to moderate (Cohen’s $d = 0.41$) for symptom reduction (as proposed here) [Jones & Sharpe (2017). Journal of Affective Disorders, 223, 175-183.]. These meta-analyses were not in pain, but the effects appear more robust in anxiety than other areas where they have been used – so on what basis do the authors expect such large effects? This question is particularly pertinent since delivery via the internet in other areas has been found consistently to result in smaller effects than delivery via other modes.

We chose to power the study on a medium effect size which would most probably be clinically significant and therefore relevant to our primary outcome. We have additionally revised the design and measures will also be taken after session 10 in the two 18 session groups (18 AMG and 18 PTG).

5. *Will the authors assess blindness for both participants and researchers at the end of the study?*

Yes, we will assess blindness, which we now clarify in the Outcome Measures subsection.

6. In ABM training studies, typically participants have not been given information about ABM given that the training is surreptitious and the mechanism is argued to be implicit. Can the authors state what is the information given to participants at baseline through the website and what is the rationale for this? Have the authors considered how this might affect the participants and the study outcomes? It may be that the lack of a rationale (and therefore motivation) to complete the task is one reason for the failure of internet-based protocols to be successful, so this might be helpful, but should be discussed.

It is explained to participants that that a person with chronic pain is likely to notice pain-related information or cues in their environment and interpret these as personally threatening. Participants are informed that ABM is a computer-based therapy used to help change these unhelpful patterns of attention which may play a role in the maintenance of chronic pain. The general process of the visual-probe task is explained (i.e., fixation cross, two stimuli, and a visual-probe requiring a response).

Participants are also informed that ABM training has no known adverse effects, is designed as an optional complement to existing therapies and does not require any changes or alterations to current treatments. It is explained however that ABMT requires commitment across several sessions.

We agree that lack of sufficient detail or rationale may understandably lead to reduced motivation or attrition. Another reason may be that participants do not understand or trust the science behind the methodology being employed and perceive the repetitive nature of the task as irrelevant. Based upon these concerns, discussions with participants in our previous ABM study¹, and current clinical practice which involves educating patients into chronic pain and the psychological interventions we use to manage it, we therefore decided to explain the rationale to participants. We now explain this in the manuscript.

7. *The authors state that they will use the “same” task at pre- and post-treatment, but it is unclear whether they mean the same stimuli. Typically, the stimuli are changed in the post-treatment assessment phase from both the pre-treatment assessment phase and training phase. Can the authors please clarify?*

We have clarified that the pre-and post-treatment assessment visual-probe tasks will use the same stimuli.

8. *I could not find sufficient details in the protocol about the actual ABM task used. As I understand it, the ABM will have 8 pairs of pictures and 8 pairs of words (16 unique combinations). For each combination, if they are presented in the four necessary combinations for the placebo, there will be 64 trials. I do not see how the ABM could take 25 minutes. Can the authors state more clearly, the number of trials, and exact parameters of ABM training in order to allow other authors to replicate. I was also confused as to whether there were 18 different combinations of the 16 pairs, or whether the training simply repeated the same stimuli. Finally, the authors state that they are using threat-related stimuli and I commend them for all the included data that they provide in supplementary materials, but I still found it hard to be clear what proportion of stimuli were pain-related, and it was only upon reading the supplementary materials that I realised that there were neutral/neutral trials included. I think there just needs to be more clarity in the paper itself.*

We now include a brief summary of the stimuli in the manuscript, although full details are provided in the much longer Supplementary Material 3 document. Each visual-probe task will include 48 unique stimuli pairs, which will each be presented 8 times (i.e., 4 times per presentation time) thus giving a total of 384 trials per task. Our piloting has confirmed each session will last approximately 35 minutes.

9. It would be excellent to see the authors state some pre-specified mediation analyses, relating to changes in biases – particularly if they have hypotheses relating to biases at 500msec vs 1250 msec.

A comprehensive Statistical Analysis Plan (SAP) will be written before study closure. Mediation models will be included in the wider plan. Our primary aim is to first establish whether our intervention has therapeutic benefits.

10. I was a little unconvinced about the planned analyses. In terms of the changes in biases and any mediation analyses, I can see that a completers' analysis makes sense, but in terms of efficacy, an intention to treat analysis would be required. I thought that the manuscript was unclear on this. Can the authors also state why they are not using a linear mixed models analysis?

Based on the reviewers' comments we have now amended our analysis plan, and will use linear mixed model with random intercepts to analyse our data. We have described in more detail our intention-to-treat analysis. As per current practice a comprehensive Statistical Analysis Plan (SAP) will be written before the data analysis commences.

Minor Issues:

1. *The authors state the results of the available meta-analyses, but omit the most recent meta-analysis (see Todd et al., (2018). Attentional bias to pain-related information: A meta-analysis of dot-probe studies. Health psychology review, 12(4), 419-436.). Further, they state that attentional biases are present "particularly at longer stimuli presentation times (i.e., > 1000 ms) associated with maintained attention as indicated in meta-analyses.21 22" This is not entirely accurate since the larger two meta-analyses of Crombez et al. (2013) and Todd et al. (2018) do not find that biases are more evident at longer presentations, and actually much of the eye tracking data supports vigilance as the primary attentional bias, where results about difficulty disengaging is less clear. So I would omit this from the sentence.*

Thank you. We have now cited the meta-analysis from Todd and colleagues and have amended the text for clarity. We clarify that only two meta-analyses provide evidence for larger effect sizes when longer stimuli presentation times are used, and that overall evidence is inconsistent.

2. *Line 43-44 on page 4 ends "another in and pain intensity". There seems to be a word missing.*

Thank you, we have now reworded.

3. *The authors suggest "The stimulus pair may be either words or images, although critically includes one threat-related and one neutral stimulus. 500 and 1250 ms presentation times are used to explore biases in initial orienting of attention and maintained attention respectively.21" The authors themselves did suggest that these durations match biases in initial orienting and maintained attention, but most researchers would suggest that this was an oversimplification. It is a strength that the authors include both, but they should definitely rephrase this. I would recommend the authors read the seminal paper by Cisler and Koster, and adopt their language about the different nature of attention biases – or, since it is not entirely relevant to the protocol paper, perhaps they could omit it altogether. (Cisler, J. M., & Koster, E. H. (2010). Clinical psychology review, 30(2), 203-216.)*

We have now amended this sentence to state that 500 and 1250 ms presentation times are the most commonly used in former visual-probe research, and we have adopted both of these in the present intervention in order to more closely compare our results to former published studies. We have also discussed evidence for the time-course of attentional bias in more detail in the Introduction.

Will adverse events be recorded?

Yes, all adverse events will be recorded in accordance with University of Southampton policies. We now state this more clearly in the manuscript.

Reviewer: 3. Reviewer Name: Dimitri Van Ryckeghem

1. *The authors describe this study as the first study investigating internet-delivered ABM in chronic pain patients. Although I agree that systematic research on ABM in chronic pain is lacking, I would like to draw the attention of the researchers to following studies. First, a study of Heathcote and colleagues (2018) provides findings of ABM in adolescents suffering chronic pain. Second, there is also an ongoing/finalized internet delivered ABM trial by Asmundson and colleagues (see Van Ryckeghem et al., 2018). Therefore, I would suggest authors do not present this study as the first*

Thank you. We have removed mention of this being the first study investigating internet-delivered ABM in chronic pain patients.

2. *The authors mention that the inclusion of a diverse chronic pain sample is strength of current study? It would be helpful to have a rationale for this suggestion. Indeed, having a very heterogeneous sample may also be a drawback for clear conclusions. Could authors better motivate their choice for a heterogeneous sample?*

We now mention in the Introduction that recruitment of a diverse sample will increase the ecological validity of results to the CMSK population. While we understand potential limitations of diverse samples, we feel this approach is especially relevant considering our intervention is being made available online.

3. *In the introduction, only the positive findings of previous ABM studies have been presented. I would suggest that authors also indicate the negative and null-findings of each of these studies. Indeed, many studies include a large number of variables of which only few are significant. Providing also an overview of these negative and null-findings results in a more complete picture of the findings of ABM research todate.*

Thank you. We now include a much more detailed description of the results of former ABM studies in the Introduction.

4. *I was surprised that the authors did not report a change in attention bias as an outcome of attention bias modification training. Why did authors not include this as an outcome? Do authors think they have sufficient power to detect a change in AB? In particular, since the correlation between AB over time is low (see Dear et al., 2011), Could authors discuss this further.*

A comprehensive Statistical Analysis Plan (SAP) will be written before study closure and this will incorporate delta attention bias. Our primary aim is to first establish whether our intervention has therapeutic benefits. We have additionally revised the design so that outcome measures in the 18 sessions groups will also be taken after session 10.

5. Could authors provide a rationale for the different age groups. Why did authors split groups based on age. Again, do authors have sufficient power to detect differences between groups? Or do they not expect a difference? What is the rationale for this particular groups? Similarly, could authors discuss the rationale for the dose? These doses are relatively large (see also doses given in anxiety and depression) compared to previous interventions.

We will analyse and report data separately for participants aged 16 – 24 and 25 – 60 based on the recent argument that adolescence now occupies a greater portion of life than ever before, up to the age of 24.² Furthermore, given the limited available evidence on the efficacy of ABMT we believe the issue of a dose depended effect is still an open question and we therefore decided to explore it. 10 vs 18 is exploratory and based on the average sessions for CBT 8 (+ 2 assessment sessions) and double the dose i.e., 16 ((+ 2 assessment sessions)

We have chosen to power this study on a medium effect size to establish treatment efficacy for the primary outcome variables of pain intensity and pain interference by considering baseline and endline measures between 10AMG and 10PTG. A comprehensive Statistical Analysis Plan (SAP) will be written before the data analysis commences.

6. In contrast to many of the previous studies, the authors have chosen for a presentation time of 500 and 1250 ms (mixed). Again, a rationale for this decision may be helpful. In particular, because based upon a different presentation time other processes may be targeted (ruminative vs early attention processes). Previous research has systematically opted for a presentation time of 500 ms. Authors may be interested in the findings of a recent review on Todd and colleagues (2018) discussing the impact of experimental procedures on the presence and magnitude of AB.

We agree with you that based upon a different presentation time, different processes may be targeted (ruminative vs early attention processes) and this is a line of inquiry we actively pursue in our lab with good results. We now explain in the manuscript that 500 and 1250 ms presentation times are most commonly used in former visual-probe research, and we have adopted both of these in the present intervention in order to more closely compare our results to former published studies. We have also discussed evidence for the time-course of attentional bias in more detail in the Introduction.

7. Authors focus on musculoskeletal pain. How will authors deal with comorbid pain (e.g., Headache, neuropathic pain,..). Will participants be excluded on this basis? Please provide clarification on the inclusion and exclusion criteria and the rationale for these criteria.

Our intervention has been designed for individuals with MSK specifically, including certain stimuli categories of relevance for this specific population (e.g., images of musculoskeletal threat). We have now added an exclusion criterion stating that individuals with a diagnosis of other form of comorbid chronic pain (e.g., chronic headache) will be excluded. It is possible that individuals with chronic headache may also find the requirements of this intervention difficult (e.g., using a computer for 40 minutes). We cannot meaningfully exclude neuropathic pain because on many occasions musculoskeletal pain is mixed and has a neuropathic component.

8. *Related to this, how will a diagnosis of any psychiatric disorder (e.g., depression and anxiety disorders are very common) will be determined (self-report, MD, ..) furthermore, what is meant with psychiatric therapy. Please clarify.*

Inclusion and exclusion criteria will be assessed via telephone interview with one of the researchers. Any diagnosis will be self-reported by the potential participant. We have now removed the criterion regarding psychiatric therapy as we believe this was redundant based on our criterion of having a diagnosis of any psychiatric disorder.

Reviewer: 4. Reviewer Name: Lucy Busija

Recruitment: Not clear how interested participants will express their interest in taking part in the study.

We now clarify that interested individuals will contact the researchers on the details provided in study advertisements.

Attrition: Please clarify the anticipated attrition rate for the study sample and how you will ensure that attrition does not compromise study power. While the use of multiple imputation to handle missing data is appropriate, multiple imputation is known to increase standard errors of estimates, with the resultant loss of power. What measures are in place to ensure that attrition does not compromise the study power at the end of follow up?

A recent review of online interventions for chronic pain reported a wide range of attrition from 4% to 54%, although for unguided trials ranged from 7.1% to 16.4%.³ We now state that a 20% oversampling will be undertaken to mitigate a loss of power from loss to follow-up.

Randomisation: The authors propose to analyse data for participants aged 16 – 24 and 25 – 60 separately. How will you ensure balanced allocation to the intervention and placebo groups in each age group?

We now clarify that blocked randomisation will be performed separately for those aged 16-24 and those aged 25-60.

Data handling: Please clarify the process of outlier removal. If any 'unusual' observations are to be removed prior to analyses, what measures are in place to ensure that the intent to treat principle is adhered to during data analysis (ie, does your process of data cleaning safeguard against the possibility of removing participants with 'unusual' response patterns from analyses, as opposed to only removing 'unusual' trials for some of the participants)?

Our data cleaning processes follows standard conventions in the literature for the visual-probe task, and is designed to remove any unusual responses on trials which could distort the attentional bias scores for the individual participant. Our former studies have typically found fewer than 4-5% of trials are removed as outliers⁴⁻⁶, and therefore we are confident that this will not compromise our intent-to-treat approach.

Data analysis: With the application of Bonferroni correction to post hoc tests, will your study have sufficient power to detect meaningful differences?

This study is powered primarily to establish treatment efficacy for the primary outcome variables of pain intensity and pain interference by considering baseline and endline measures between

10AMG and 10PTG. As per current practice a comprehensive Statistical Analysis Plan (SAP) will be written before the data analysis commences, and we will carefully inspect and comment on statistical power in all our analyses.

Throughout text: 'randomised control trial' should be 'randomised controlled trial'; please check the manuscript carefully and correct the terminology.

We have now changed all instances to randomised controlled trials.

VERSION 2 – REVIEW

REVIEWER	Lauren Heathcote Stanford University School of Medicine
REVIEW RETURNED	05-Aug-2019

GENERAL COMMENTS	The authors have addressed my comments and I support publication of this protocol.
--

REVIEWER	Louise Sharpe School of Psychology University of Sydney NSW Australia 2006
REVIEW RETURNED	06-Aug-2019

GENERAL COMMENTS	The authors have made appropriate changes to the manuscript. This study protocol describes a well controlled study that is needed in this area.
---

REVIEWER	Dimitri Van Ryckeghem Maastricht University
REVIEW RETURNED	29-Aug-2019

GENERAL COMMENTS	I was glad to see that the authors have addressed several of the concerns of the reviewers. Yet, from my side there remain some issues that deserve further attention or are not completely clear yet. I specify these below: 1) Although the authors provide some rationale on why they take look at age and dose as important factors to explain differences in the effect of ABM, it is still largely lacking in the introduction. Could authors add the rationale more in the text. Related to this, I am concerned about the power in the current study. Could authors clearly indicate how many participants they will recruit in each subgroup and whether this is sufficient to detect small to moderate effects in each subgroup. Indeed, if ABM does not work in adolescence, authors should still have sufficient power to detect ABM effects in adulthood (a similar rationale accounts for dose). Could authors be clear on the n (power) to detect small to moderate effects in each subgroup (certainly as it is suggested that subgroups may be analyzed separately). 2) The authors have now better indicated how the statistical plan might look like. I do however think that the current protocol would benefit from the inclusion of a more detailed statistical plan
--

	already. The authors refer to the fact that they will develop a full statistical Analysis Plan prior to trial closure. I would propose this plan is already included here (maybe as supplementary materials). Yet, I leave the final decision to the editor whether this is needed at this stage. 3) The authors refer to the BPI as measure for primary outcome. Could authors also indicate which subscale/item of the BPI they will use to assess the pain intensity, as different authors have used different items/subscales in the past. 4) I have missed this during the first revision, but it is unclear to me why authors use a q or p as probe if localizing the probe is the task. Indeed, p and q are normally used when identification of the probe is the task, not localization. In the case of localization the probe is normally always the same. Personally, I would think that identification is the better option here (certainly for AB assessment). This is less of importance during the training. Could authors explain why they use this approach? 5) I think it should be clear when people will be considered not to adhere instructions and excluded from the trial. What will happen with people that became aware of the allocation, ... Please clarify in the text and spell out all exclusion criteria clearly. 6) Finally, if I understood correctly, different words will be used during different training sessions and the order of the sessions is fixed. I wonder why authors have not opted to randomize this (with the limitation that both a ABM with pictures and a ABM with words is presented during each week)? This may avoid the problem that dose is confounded with the type of stimuli used during the last 8 sessions.
--	---

REVIEWER	Lucy Busija Monash University, Australia
REVIEW RETURNED	11-Aug-2019

GENERAL COMMENTS	Comments previously raised by reviewers appear to have been addressed satisfactorily. In light of new information provided and changes to the analytical model, I have a few additional queries that require to be addressed: Description of power analysis on page 8: this section describes power for two different effect sizes. However, it is not made clear which effect size is of primary interest or how each of the two effect sizes relates to the hypotheses and proposed analyses. Also, it is not clearly stated how many individuals you will need to recruit in each age group to be able to have sufficient power to test your primary hypotheses. Quantitative analyses: There is insufficient information about the proposed mixed model – which variables will be include in the model and which variables will be modelled as fixed or random effects? Missing data handling: please also clarify what assumption is being made regarding the expected missing data mechanism(s) in your study (MAR, MCAR, MNAR). Also, I am not convinced by the proposed approach to sensitivity analyses for missing data. Both mixed model and multiple imputation make the same assumption regarding missing data mechanism (MAR). To truly assess ‘sensitivity’ of results to missing data mechanisms (which in practice are not known), a more rigorous approach would be to utilise approaches to missing data handling that make different
--

	assumptions. The authors may wish to consult work by Ian White on sensitivity analyses for missing data (BMJ 2011;342:d40).
--	---

VERSION 2 – AUTHOR RESPONSE

Reviewer: 1. Reviewer Name: Lauren Heathcote

The authors have addressed my comments and I support publication of this protocol.

Thank you very much.

Reviewer: 2. Reviewer Name: Louise Sharpe

The authors have made appropriate changes to the manuscript. This study protocol describes a well controlled study that is needed in this area.

Thank you very much.

Reviewer: 3. Reviewer Name: Dimitri Van Ryckeghem

1. Although the authors provide some rationale on why they take look at age and dose as important factors to explain differences in the effect of ABM, it is still largely lacking in the introduction. Could authors add the rationale more in the text.

We have now provided more rationale in the Introduction for exploring dose and age.

Related to this, I am concerned about the power in the current study. Could authors clearly indicate how many participants they will recruit in each subgroup and whether this is sufficient to detect small to moderate effects in each subgroup. Indeed, if ABM does not work in adolescence, authors should still have sufficient power to detect ABM effects in adulthood (a similar rationale accounts for dose). Could authors be clear on the n (power) to detect small to moderate effects in each subgroup (certainly as it is suggested that subgroups may be analyzed separately).

The authors have now better indicated how the statistical plan might look like. I do however think that the current protocol would benefit from the inclusion of a more detailed statistical plan already. The authors refer to the fact that they will develop a full statistical Analysis Plan prior to trial closure. I would propose this plan is already included here (maybe as supplementary materials). Yet, I leave the final decision to the editor whether this is needed at this stage.

Thank you. We now explicitly state the number to be randomised (N= 100). We have included Age group as a subgroup analysis by considering the Age by Treatment interaction i.e. to use all available data and not to directly subset as the main analytical approach for Age. This will be a better analysis strategy as the distribution of Age cannot be directly controlled at recruitment. We have extended the statistical analysis plan so all main analyses are evident.

3. The authors refer to the BPI as measure for primary outcome. Could authors also indicate which subscale/item of the BPI they will use to assess the pain intensity, as different authors have used different items/subscales in the past.

After careful consideration the pain interference subscale will be used as the primary outcome measure. In addition, we will follow recommendations outlined in the Brief Pain Inventory User Guide¹ that all four subscales are used (i.e., 'worst', 'least', 'average' and 'now' pain), and which is supported by IMMPACT recommendations for assessing pain in clinical trials.² We now clarify this in the manuscript.

4. I have missed this during the first revision, but it is unclear to me why authors use a q or p as probe if localizing the probe is the task. Indeed, p and q are normally used when identification of the probe is the task, not localization. In the case of localization the probe is normally always the same. Personally, I would think that identification is the better option here (certainly for AB assessment). This is less of importance during the training. Could authors explain why they use this approach?

We opted to use the probe-classification version of the visual-probe task (i.e., P & Q) versus the probe-position version (i.e., a single dot) as this version has been used in all former published papers exploring attentional bias modification in chronic pain,³⁻⁶ and thus allows for better comparison of results. It has been argued that the probe-classification version encourages a more even monitoring of the visual-probe display than the probe-position version, as for the latter participants can adopt biased monitoring strategies, favouring one particular region over the other.⁷ As any such biases in monitoring may affect training, we use the probe-classification version of the visual-probe task.

5. I think it should be clear when people will be considered not to adhere instructions and excluded from the trial. What will happen with people that became aware of the allocation, ... Please clarify in the text and spell out all exclusion criteria clearly.

As outlined in the Quantitative Analysis subsection, data analysis will be performed on an intention-to-treat (ITT) basis to provide an unbiased estimate of the treatment effect.⁸ However we will also perform a per protocol analyses (PPA) comprising those correctly randomized, without protocol violations, and providing complete data on the primary outcome measure. While the latter provides a lower level of evidence, it better reflects treatment effects when taken in an optimal manner.⁹ We now provide more details on protocol violations in the manuscript.

We will assess at endline whether participants became aware at any point which group they had been allocated to. Should this prove to be the case, where possible we will perform moderator analyses to explore the impact on study outcomes. Limitations and potential biases that may have been introduced by awareness of group allocation will also discussed in the final manuscript.¹⁰

6. Finally, if I understood correctly, different words will be used during different training sessions and the order of the sessions is fixed. I wonder why authors have not opted to randomize this (with the limitation that both a ABM with pictures and a ABM with words is presented during each week)? This may avoid the problem that dose is confounded with the type of stimuli used during the last 8 sessions.

Different words and images are used during each training session and the order of the sessions is fixed. Although we understand the potential benefits of randomising training session order, we have

taken great care to ensure that stimuli do not differ in any substantial way between sessions. For each category of stimuli (i.e., sensory-pain words, facial expressions of pain) the threatening stimuli were first randomly allocated to each of the eight linguistic or pictorial training sessions. The order of each training linguistic and pictorial session was then randomised. Considering this, and the fact we will be running per protocol analysis in addition to intention-to-treat analysis, we have decided not to modify our programmed intervention at this stage.

Reviewer: 4. Reviewer Name: Lucy Busija

Description of power analysis on page 8: this section describes power for two different effect sizes. However, it is not made clear which effect size is of primary interest or how each of the two effect sizes relates to the hypotheses and proposed analyses. Also, it is not clearly stated how many individuals you will need to recruit in each age group to be able to have sufficient power to test your primary hypotheses.

Thank you. We have retained the most important power calculation relating to the endpoint and have now clearly stated the primary analysis and endpoint. We have altered the analytical rationale for age group and will assess the effects of age in a subgroup analysis using an Age by Treatment interaction. It is conceded the numbers recruited in each age group is itself a random variable and assessing the effect of Age group as a moderator in this way would be a better approach rather than subsetting as a direct analysis.

Quantitative analyses: There is insufficient information about the proposed mixed model – which variables will be include in the model and which variables will be modelled as fixed or random effects?

We now provide more information in the Quantitative Analysis section.

Missing data handling: please also clarify what assumption is being made regarding the expected missing data mechanism(s) in your study (MAR, MCAR, MNAR). Also, I am not convinced by the proposed approach to sensitivity analyses for missing data. Both mixed model and multiple imputation make the same assumption regarding missing data mechanism (MAR). To truly assess 'sensitivity' of results to missing data mechanisms (which in practice are not known), a more rigorous approach would be to utilise approaches to missing data handling that make different assumptions. The authors may wish to consult work by Ian White on sensitivity analyses for missing data (BMJ 2011;342:d40).

The reviewer makes a very pertinent point re sensitivity of findings to missing data mechanisms. The work of White¹¹ and his group of colleagues¹² has provided some good consideration and, consequently, we have revised the sensitivity to missing data mechanisms in this protocol. Specifically, we will use the strategy alluded to a briefly discussed by Morris et al, whereby imputed values are altered by a randomised group dependent value prior to analysis. We will run this sensitivity analysis over a factorial arrangement of group dependent values constraining altered imputed values to be in the valid response range. The case of the group dependent values being equal to zero will correspond to multiple imputation under MAR and non-zero values would reflect deviations away from this assumption.

References

1. Cleeland CS. The brief pain inventory: user guide. Houston, TX: The University of Texas MD Anderson Cancer Center 2009:1-11.
2. Dworkin RH, Turk DC, Wyrwich KW, et al. Interpreting the clinical importance of treatment outcomes in chronic pain clinical trials: IMMPACT recommendations. *J Pain* 2008;**9**(2):105-21.
3. Schoth DE, Delgado Nunes V, Lioffi C. Attentional bias towards pain-related information in chronic pain; a meta-analysis of visual-probe investigations. *Clin Psychol Rev* 2012;**32**(1):13 - 25 doi: 10.1016/j.cpr.2011.09.004.
4. Sharpe L, Ianiello M, Dear BF, et al. Is there a potential role for attention bias modification in pain patients? Results of 2 randomised, controlled trials. *Pain* 2012;**153**(3):722-31 doi: 10.1016/j.pain.2011.12.014.
5. Carleton RN, Richter AA, Asmundson GJG. Attention Modification in Persons with Fibromyalgia: A Double Blind, Randomized Clinical Trial. *Cogn Behav Ther* 2011;**40**(4):279-90 doi: 10.1080/16506073.2011.616218.
6. Heathcote LC, Jacobs K, Van Ryckeghem DML, et al. Attention bias modification training for adolescents with chronic pain: a randomized placebo-controlled trial. *Pain* 2018;**159**(2):239 - 51 doi: 10.1097/j.pain.0000000000001084.
7. Mogg K, & Bradley, B. P. Some methodological issues in assessing attentional biases for threatening faces in anxiety: a replication study using a modified version of the probe detection task. *Behav Res Ther* 1999;**37**(6):595 - 604.
8. Gupta SK. Intention-to-treat concept: a review. *Perspectives in Clinical Research* 2011;**2**(3):109 doi: 10.4103/2229-3485.83221.
9. Sedgwick P. Intention to treat analysis versus per protocol analysis of trial data. *BMJ* 2015;**350**:h681.
10. Karanicolos PJ, Farrokhyar F, Bhandari M. Blinding: Who, what, when, why, how? *Can J Surg* 2010;**53**(5):345.
11. White IR, Horton NJ, Carpenter J, et al. Strategy for intention to treat analysis in randomised trials with missing outcome data. *BMJ* 2011;**342**:d40.
12. Morris TP, Kahan BC, White IR. Choosing sensitivity analyses for randomised trials: principles. *BMC Med Res Methodol* 2014;**14**(1):11.

VERSION 3 – REVIEW

REVIEWER	Dimitri Van Ryckeghem Maastricht University
REVIEW RETURNED	26-Dec-2019

GENERAL COMMENTS	The authors have done a thorough revision of the manuscript and addressed the majority of my questions. Could authors check the manuscript and report consistently that they are using probe-classification version of the visual-probe task and describe the task as such. It is currently still stated that "Participants indicate the location of the probe as quickly and as accurately as possible via corresponding keys on their keyboard."
--

REVIEWER	Lucy Busija Monash University, Australia
REVIEW RETURNED	18-Dec-2019

GENERAL COMMENTS	The authors have addressed my comments very thoroughly and thoughtfully and I have no further comments or suggestions regarding this manuscript.
--

VERSION 3 – AUTHOR RESPONSE

Reviewer 3. Reviewer Name: Dimitri Van Ryckeghem

The authors have done a thorough revision of the manuscript and addressed the majority of my questions. Could authors check the manuscript and report consistently that they are using probe-classification version of the visual-probe task and describe the task as such. It is currently still stated that "Participants indicate the location of the probe as quickly and as accurately as possible via corresponding keys on their keyboard."

Thank you very much. We now report consistently that the probe-classification version of the visual-probe task is used, and have amended the description in the Method accordingly.

Reviewer 4. Reviewer Name: Lucy Busija

The authors have addressed my comments very thoroughly and thoughtfully and I have no further comments or suggestions regarding this manuscript.

Thank you very much.